# Comparative Analysis of Radical Adduct Formation (RAF) Products and Antioxidant Pathways between Myricetin-3-*O*-Galactoside and Myricetin Aglycone

**DOI:** 10.3390/molecules24152769

**Published:** 2019-07-30

**Authors:** Xican Li, Xiaojian Ouyang, Minshi Liang, Dongfeng Chen

**Affiliations:** 1Innovative Research & Development Laboratory of TCM of Guangdong Province, University of Chinese Medicine, Guangzhou 510006, China; 2School of Chinese Herbal Medicine; Guangzhou University of Chinese Medicine, Guangzhou 510006, China; 3School of Basic Medical Science, Guangzhou University of Chinese Medicine, Guangzhou 510006, China; 4The Research Center of Integrative Medicine, Guangzhou University of Chinese Medicine, Guangzhou 510006, China

**Keywords:** myricetin-3-*O*-galactoside, myricetin, 3-*O*-galactosylation, antioxidant pathway, radical adduct formation

## Abstract

The biological process, 3-*O*-galactosylation, is important in plant cells. To understand the mechanism of the reduction of flavonol antioxidative activity by 3-*O*-galactosylation, myricetin-3-*O*-galactoside (M3OGa) and myricetin aglycone were each incubated with 2 mol α,α-diphenyl-β-picrylhydrazyl radical (DPPH^•^) and subsequently comparatively analyzed for radical adduct formation (RAF) products using ultra-performance liquid chromatography coupled with electrospray ionization quadrupole time-of-flight tandem mass spectrometry (UPLC-ESI-Q-TOF-MS) technology. The analyses revealed that M3OGa afforded an M3OGa–DPPH adduct (*m*/*z* 873.1573) and an M3OGa–M3OGa dimer (*m*/*z* 958.1620). Similarly, myricetin yielded a myricetin–DPPH adduct (*m*/*z* 711.1039) and a myricetin–myricetin dimer (*m*/*z* 634.0544). Subsequently, M3OGa and myricetin were compared using three redox-dependent antioxidant analyses, including DPPH^•^-trapping analysis, 2-phenyl-4,4,5,5-tetramethylimidazoline-1-oxyl 3-oxide radical (PTIO^•^)-trapping analysis, and ^•^O_2_ inhibition analysis. In the three analyses, M3OGa always possessed higher IC_50_ values than those of myricetin. Conclusively, M3OGa and its myricetin aglycone could trap the free radical via a chain reaction comprising of a propagation step and a termination step. At the propagation step, both M3OGa and myricetin could trap radicals through redox-dependent antioxidant pathways. The 3-*O*-galactosylation process, however, could limit these pathways; thus, M3OGa is an inferior antioxidant compared to its myricetin aglycone. Nevertheless, 3-*O*-galactosylation has a negligible effect on the termination step. This 3-*O*-galactosylation effect has provided novel evidence that the difference in the antioxidative activities of phytophenols exists at the propagation step rather than the termination step.

## 1. Introduction

Galactosylation is an important biological process in cellular metabolism [1,2] that is catalyzed by certain enzymes, such as bovine β (1,4)-galactosyltransferase [2], Sb3GT1 (UGT78B4) [3], AgUCGalT1 [4], and α-1,3-galactosyltransferase (α3GalT) [5]. Through galactosylation, plant cells can link a galactose residue to the 3-*O* atom in flavonol. This process is termed “3-*O*-galactosylation” [6]. The 3-*O*-galactosylation process results in extensive coexistence of flavonol 3-*O*-galactoside and flavonol aglycone in the same plant. Thus far, at least six pairs of flavonol 3-*O*-galactoside and flavonol aglycone have been found in identical plants (Appendix A) [7,8,9,10,11,12].

Flavonol is well known as an effective natural antioxidant. Experimental [13,14,15] and theoretical studies [16,17,18] have indicated that the antioxidant activity of flavonol is closely associated with the presence of 3-OH. Accordingly, 3-*O*-galactosylation is believed to reduce the antioxidant activity of flavonol, although no study on the mechanism of this reduction has been conducted.

Consequently, myricetin-3-*O*-galactoside (M3OGa) and its myricetin aglycone were selected as the representatives for the comparative study. As shown in Figure 1A, M3OGa bears a β-galactose residue at its 3-O position; thus, it can be regarded as the 3-*O*-galactosylation derivative of myricetin. If any difference in their antioxidant activities exists, it can be attributed to 3-*O*-galactosylation. Recently, M3OGa has been reported to coexist with its myricetin aglycone in white myrtle [11] and Nelumbo nucifera [12]. Their coexistence in the same plant has actually enhanced the comparability and biologically relevance of this comparative study.

In the comparative study, the final products of the interaction of M3OGa and myricetin aglycone with α,α-diphenyl-β-picrylhydrazyl radical (DPPH^•^) were severally analyzed using leading-edge ultra-performance liquid chromatography coupled with electrospray ionization quadrupole time-of-flight tandem mass spectrometry (UPLC-ESI-Q-TOF-MS) technology to test the possibility of radical adduct formation (RAF). The high resolution of the Q-TOF-MS technology ensures the reliability of the chemical analysis. Based on the RAF product analysis, M3OGa and myricetin were further investigated for their antioxidant pathways using relevant chemical approaches. Expectedly, the series of investigative experiments will provide profound knowledge on the mechanism of the reduction of the antioxidant activity of flavonol by 3-*O*-galactosylation.

In addition, the understanding of the 3-*O*-galactosylation process is expected to be of benefit to other types of 3-*O*-glycosylation processes, such as 3-*O*-glucosylation, 3-*O*-rhamnglycosylation, and 3-*O*-arabinosylation. This is because these 3-*O*-glycosylation processes are essentially not different from 3-*O*-galactosylation, and flavonoid-3-*O*-glycosides are present in plants (e.g., myricetin-3-*O*-glucoside, myricetin-3-*O*-rhamnoside, and myricetin-3-*O*-arabinoside). From an antioxidant chemistry viewpoint, flavonol (or its glucoside) has the same antioxidant pathways as those of other phytophenols [19,20,21,22]. Thus, the analysis of the RAF products, based on the UPLC-ESI-Q-TOF-MS technology, will provide novel and reliable insights on the antioxidant chemistry of all types of phytophenols, especially flavonoid 3-*O*-glucosides (e.g., isorhamnetin 3-*O*-galactoside [7], hyperin [8], trifolin [9], and syringetin 3-*O*-galactoside [10]) and anthocyanin 3-*O*-galactosides (e.g., cyanidin-3-*O*-galactoside [23] and delphinidin-3-*O*-galactoside [24]). 

## 2. Results and Discussion

According to previous reports [25,26], phytophenol antioxidants can afford final products through the RAF reaction when they interact with free radicals [27,28,29]. Thus, the RAF product analysis may be a potential tool for exploring antioxidant chemistry. The present study, however, used cut-edging UPLC-ESI-Q-TOF-MS technology to analyze the RAF products of M3OGa and myricetin interacting with 2 mol DPPH^•^ radical.

After treatment with 2 mol DPPH^•^, 1 mol of M3OGa yielded a chromatographic peak at a retention time (Rt) of 2.486 min in the analysis. Moreover, the peak further afforded an *m*/*z* 873.1573 molecular ion peak in the MS spectra (Figure 2D). This value (*m*/*z* 873.1573) represented the loss of exactly one H atom (*m*/*z* 1.0091 [30]) as compared to the sum of the molecular weights of M3OGa (*m*/*z* 480.0858) and DPPH^•^ (*m*/*z* 394.0806, Appendix A). Thereby, M3OGa may combine with DPPH to generate an H atom and an adduct. The adduct can be further verified by the two characteristic fragments, *m*/*z* 196 and 226 (Figure 2F). Our previous studies indicated that DPPH^•^ or the DPPH adduct always affords *m*/*z* 196 and 226 fragments in the negative ion model MS spectra [13,27,31,32,33,34]. Considering that the above adduct is indeed M3OGa–DPPH, the relative deviation between the experimental value (*m*/*z* 872.1482) and the calculated molecular weight value (M.W. 872.1535) was only 6.1 × 10^−6^. Therefore, an M3OGa–DPPH adduct was definitively formed when M3OGa interacted with DPPH^•^. Based on previously reported studies [13,27,31,32,33,34], the MS spectra of the M3OGa–DPPH adduct could be elucidated, as shown in Figure 3A. 

Besides the adduct, a flavonol dimer was formed from the interaction with the free radical [35]. As illustrated in Figure 2G–H, M3OGa (*m*/*z* 480.0858) afforded an MS peak with *m*/*z* 958.1620 at an Rt of 1.374 min. This strongly indicates that two M3OGa molecules dimerized into M3OGa–M3OGa. During the dimerization process, two H atoms may have been lost via H atom transfer (HAT) pathways (see below). 

Thus, it is clear that M3OGa treated with 2 mol DPPH^•^ can yield an M3OGa–DPPH adduct and an M3OGa–M3OGa dimer. The adduct and dimer are the two main products of the RAF reactions. Similarly, myricetin could also produce a myricetin–DPPH adduct and myricetin–myricetin dimers (Figure 2M–P). The proposed RAF products of the myricetin reaction with DPPH^•^ and their MS elucidations are shown in Figure 4. The generation of two types of RAF products, however, could offer further insights into the antioxidant chemistry of M3OGa and myricetin. Considering that the reaction proceeds via a concerted pathway, excessive DPPH^•^ (2 mol) will synchronously link the M3OGa molecule to produce an M3OGa–DPPH adduct and no M3OGa–M3OGa dimer. Conversely, considering that the reaction is ion-mediated (rather than radical-mediated), M3OGa may be converted into M3OGa (or M3OGa^Θ^). M3OGa (or M3OGa^Θ^) ions cannot link with each other. The co-existence of the M3OGa–DPPH adduct and M3OGa–M3OGa dimer in the product mixture strongly suggests that the DPPH-trapping reaction of M3OGa is a radical-mediated stepwise reaction. The reactivity of the free radicals is so high that their stepwise reaction becomes a chain reaction. This agrees with the previous literatures [36,37]. Furthermore, the literature has indicated that radical-trapping chain reactions can be fulfilled via two steps (propagation and termination). In accordance with this, the DPPH^•^-trapping reaction of M3OGa can be proposed, as shown in Figure 5.

As shown in Figure 5, the intermediate radicals play a key role in the radical chain reaction. Intermediate radicals were formed when DPPH^•^ accepted an H atom (H^•^) and the M3OGa antioxidant donated an H atom (H^•^). Several successful pathways have been reported to achieve this change. The first pathway is called the HAT pathway, where an H atom as one unit is directly transferred from an antioxidant molecule to a free radical. In fact, HAT has been suggested as the main pathway of the DPPH^•^-trapping reaction [38]. The results in Appendix A show that M3OGa and myricetin increased their DPPH^•^-trapping abilities in a dose-dependent manner, implying that both M3OGa and myricetin might have performed their antioxidant activities via the HAT pathway. This may responsible for the above observation of *m*/*z* 1.0091. 

Besides the HAT pathway, other pathways may also occur, including mere electron transfer (ET) and ET plus proton transfer (PT). Furthermore, the ET plus proton transfer pathway can be classified into four subtypes, i.e., proton loss single electron transfer (SPLET) [19,37,39,40,41], sequential electron-proton transfer (SEPT) [42], proton-coupled electron transfer (PCET) [19,37,40,41,43], and concerted proton–electron transfer (CPET) [44]. To test the possibility of the ET pathway, M3OGa and myricetin were determined using PTIO^•^-trapping at pH 4.5. Cyclic voltammetry evidence indicated that PTIO^•^-trapping at pH 4.5 is an ET-mediated process [45]. The results in Appendix A show that M3OGa and myricetin effectively trapped the PTIO^•^ radical at pH 4.5, implying that they were able to undergo ET to trap the radical during the propagation step. At pH 7.4, M3OGa and myricetin could also dose-dependently trap the PTIO^•^ radical (Appendix A). However, their IC_50_ values were lower than those at pH 4.5 (Table 1). These effects of pH suggest that the H^+^-transfer pathway might mediate PTIO^•^-trapping during the propagation step. In short, the PTIO^•^-trapping analyses at both pH 4.5 and 7.4 indicate that ET *plus* PT serve as the antioxidative pathway of M3OGa and myricetin in physiological aqueous solution.

The above two free radicals, PTIO^•^ and DPPH^•^, actually cannot be found in cells, and they are merely chemical probes for antioxidant study. Dissimilar to the case with the two free radicals, the superoxide radical (^•^O_2_^−^) occurs in cells and is a member of the reactive oxygen species (ROS) family. Previous studies have suggested that the scavenging of the superoxide radical is involved in ET and PT [46,47,48,49,50]; this was further supported by our observations that M3OGa and myricetin successfully scavenged ^•^O_2_^−^ in a concentration-dependent manner (Appendix A). 

Moreover, all these antioxidant pathways involve ET, which is the basis of the redox reaction. Therefore, these pathways could be collectively termed as redox-dependent pathways. The redox-dependent pathways, however, occur predominantly in the propagation step (Figure 5). In these redox-dependent antioxidant analyses, M3OGa always afforded higher IC_50_ values (μM) than those of myricetin (Table 1), suggesting that the 3-*O*-galactosylation process limits the redox-dependent pathways in the propagation step. Such a detrimental effect may be attributed to the fact that 3-*O*-galactosylation has reduced the amount of phenolic –OH. In addition, the conformation analysis revealed that the 3-*O*-galactosylation process introduces a bulky β-galactose residue to twist the planar molecule (Figure 6). The breakage of the planarity has been reported to weaken the π-π conjugation [51] and further reduce the antioxidant activity [52]. This makes M3OGa an inferior antioxidant compared to its myricetin aglycone. 

Nevertheless, M3OGa and myricetin have previously been shown to similarly produce a dimer product and an adduct product when interacting with the DPPH radical. This similarity indicates that the 3-*O*-galactosylation process could slightly affect the RAF pathways and that it plays a negligible role in the termination step. In other words, 3-*O*-galactosylation, as one of the structural factors in phytophenols, can only undergo the propagation step to exert its effect.

Besides the 3-*O*-galactosylation process, other structural factors may also affect the antioxidant activity of phytophenols, such as *C*-glycosidation, glucuronidation, isoprenylation, methylation, geometrical configuration, and *p*-coumaroylation [32,52,53,54,55,56]. However, the effects of all these structural factors were thought to be concentrated in the propagation step. 

As reported in the literature [19,37], during the chain reaction, the propagation step proceeds via the HAT (or ET *plus* PT) pathway to generate intermediate radicals. The previous literature has suggested that if undergoing the HAT pathway, phytophenols may require 77.0–86.7 kcal/mol BDE (bond dissociation enthalpy or bond dissociation energy) to produce intermediate radicals [19,37]. However, if undergoing the ET plus PT pathway, phytophenols may require ionization energy (IE) and vertical electron affinity (EA) [57,58]. For instance, the IE and EA values of the xanthones family were 100 kcal/mol and 40 kcal/mol, respectively [59]. This implies that the generation of radicals is always difficult, regardless of the antioxidant pathway employed.

The termination step, however, occurs predominantly via the RAF pathway, a bonding reaction. Conventionally, the bonding reaction is exothermic; thus, the termination step is spontaneous and rapid. Therefore, the rate at which the antioxidant traps the free radical depends on the propagation step, and the propagation step is the rate-determining step of the whole radical trapping process of phytophenols. The structural factors (such as skeleton or substitute) can classify phytophenols into different types or subtypes; however, they can only impose their effects during the propagation step and not the termination step. Conversely, phytophenols have essentially no difference in their antioxidant chemistry, and their antioxidant reactions are radical-mediated chain reactions [19,20,21,22]. Thus, it can be inferred that for the phytophenols family, the differentiation of antioxidant activities may occur at the propagation step rather than the termination step. Generally, this was in agreement with our novel evidence. 

## 3. Materials and Methods

### 3.1. Chemicals

M3OGa (C_21_H_20_O_13_, CAS number: 15648-86-9, M.W.: 480.4, purity: 98%, Appendix A) was obtained from BioBioPha Co., Ltd. (Kunming, China); myricetin (C_15_H_10_O_8_, CAS number: 529-44-2, M.W.: 318.2, purity: 98%, Appendix A) was obtained from Chengdu Biopurify Phytochemicals Ltd. (Chengdu, China). Pyrogallol and (±)-6-hydroxyl-2,5,7,8-tetramethylchromane-2-carboxylic acid (Trolox) were obtained from Sigma-Aldrich (Shanghai, China). The α,α-Diphenyl-β-picrylhydrazyl radical (DPPH^•^, C_18_H_12_N_5_O_6_) was obtained from Aladdin Chemical Ltd. (Shanghai, China). The 2-phenyl-4,4,5,5-tetramethylimidazoline-1-oxyl-3-oxide radical (PTIO^•^) was obtained from TCI Chemical Co. (Shanghai, China). Tris-hydroxymethyl amino methane (Tris) was obtained from Dingguo Biotechnology Ltd. (Beijing, China). Methanol and the other reagents were purchased from Guangdong Guanghua Chemical Plants Co., Ltd. (Shantou, China). 

### 3.2. UPLC-ESI-Q-TOF-MS Analysis of DPPH^•^ Reaction Products with M3OGa and Myricetin

The reaction of DPPH^•^ with M3OGa and its myricetin aglycone proceeded under the conditions described in a previous paper [60]. In brief, a methanol solution of M3OGa was mixed with a methanol DPPH^•^ solution with a molar ratio of 1:2, and the resulting mixture was incubated for 24 h at room temperature. Subsequently, the product was passed through a 0.22 μm filter for UPLC-ESI-Q-TOF-MS analysis.

The UPLC-ESI-Q-TOF-MS analysis was based on the method described in our previous study [61]. The UPLC-ESI-Q-TOF-MS analysis system was equipped with a Phenomenex Luna C_18_ column (2.1 mm inner diameter × 100 mm, 1.6 μm, Phenomenex Inc., Torrance, CA, USA). The mobile phase was employed for the elution of the system and consisted of a mixture of methanol (phase A) and 0.1% formic acid water (phase B). The column was eluted at a flow rate of 0.2 mL/min with the following gradient elution program: 0–2 min, maintain 30% B; 2–10 min, 30–0% B; 10–12 min, 0–30% B. The sample injection volume was set at 3 μL for the separation of the different components. The Q-TOF-MS analysis was performed on a Triple TOF 5600*^plus^* mass spectrometer (AB SCIEX, Framingham, MA, USA) equipped with an ESI source, which was run in the negative ionization mode. The scan range was set at 100–2000 Da. The system was run with the following parameters: ion spray voltage, −4500 V; ion source heater temperature, 550 °C; curtain gas pressure (CUR, N_2_), 30 psi; nebulizing gas pressure (GS1, Air), 50 psi; Tis gas pressure (GS2, Air), 50 psi. The declustering potential (DP) was set at −100 V, whereas the collision energy (CE) was set at −45 V with a collision energy spread (CES) of 15 V. For comparison, the myricetin mixed with methanol DPPH^•^ solution was also analyzed under the above UPLC–ESI–Q–TOF–MS conditions.

### 3.3. DPPH^•^ Radical-Trapping Analysis 

The DPPH^•^ radical-trapping was determined as previously described [62]. Briefly, 80 μL of DPPH^•^ solution (0.1 mol/L) was mixed with methanolic sample solutions at the indicated concentration (*x* = 0–10 μL, 0.05 mg/mL). The mixture was maintained at room temperature, and the absorbance was measured at 519 nm on a microplate reader. The percentage of DPPH^•^ scavenging activity was calculated as follows (Equation (1)):(1)$$\mathrm{Inhibition}\;\%\;=\;\frac{A_{0}-A}{A_{0}}\;\times \;100\%,$$
where *A*_0_ is the absorbance of the control without the sample, and *A* is the absorbance of the reaction mixture with the sample.

### 3.4. PTIO^•^-Trapping Spectrophotometric Analysis 

The PTIO^•^-trapping analyses (at pH 4.5 or pH 7.4) were conducted based on our previously reported method [63]. In brief, the test sample solution (*x* = 4–20 μL, 0.25 mg/mL) was added to (20 − *x*) μL of 95% ethanol, followed by 80 μL of an aqueous PTIO^•^ solution. The aqueous PTIO^•^ solution was prepared using a phosphate-buffer solution (0.1 mM, pH 4.5 or pH 7.4). The mixture was maintained at 30 ℃ for 1 h, and the absorbance was subsequently measured at 560 nm using a microplate reader. The PTIO^•^ percentage inhibition was calculated based on the formula presented in Section 3.3.

### 3.5. Superoxide Anion (^•^O_2_^−^)-Scavenging Spectrophotometric Analysis (Pyrogallol Autoxidation Method) 

The superoxide anion (^•^O_2_^−^)-trapping activity was determined using a method previously developed in our laboratory [64]. Briefly, a 5–25 μL sample solution (0.5 mg/mL) was added to a 0.05 M Tris–HCl buffer (pH 7.4) containing Na_2_EDTA (1 mM), and the total volume was adjusted to 980 μL using a buffer. Pyrogallol (20 μL) (1,2,3-trihydroxylbenzene) solution (60 mM in 1 mM HCl) was added to the sample, and the resulting mixture was vigorously agitated before being analyzed at 325 nm every 30 s for 5 min. The ^•^O_2_^−^ radical-trapping ability was calculated as follows (Equation (2)):(2)$$\mathrm{Inhibition}\;\text{\%}\;=\;\frac{\left(\frac{\Delta A_{325\mathrm{nm},\mathrm{control}}}{T}\right)\;-\;\left(\frac{\Delta A_{325\mathrm{nm},\mathrm{sample}}}{T}\right)\;}{\left(\frac{\Delta A_{325\mathrm{nm},\mathrm{control}}}{T}\right)\;}\times \;100\%,$$
where Δ*A*_325 nm, control_ is the increase in the *A*_325 nm_ value of the mixture without the sample; Δ*A*_325 nm, sample_ is the increase in the *A*_325 nm_ value of the mixture with the sample; *T* is the time required for the determination (5 min in this case).

### 3.6. Preferential Conformation Analysis by Computational Chemistry and Molecular Weight Calculation

The preferential conformation was analyzed based on force fields by computational chemistry. In brief, the energy minimization of both M3OGa and myricetin were respectively calculated through molecular mechanics II (MM2) using the Chem3D Pro14.0 program (PerkinElmer, Waltham, MA, USA) [65,66,67]. The preferential conformation was expressed using the molecular models in Figure 6A–D.

The Q-TOF-MS analysis is characterized by highly accurate *m*/*z* values, particularly molecular weights. The molecular weight calculation based on the formula is vital for comparison with the *m*/*z* values from the Q-TOF-MS analysis. In the present study, the molecular weight calculations of M3OGa and myricetin were conducted based on the accurate relative atomic masses. The relative atomic masses of C, H, O, and N were 12.0000, 1.007825, 15.994915, and 14.003074, respectively [30].

### 3.7. Statistical Analysis

The results were reported as the mean ± SD of three independent measurements, the IC_50_ values were calculated by linear regression analysis, and independent-sample *t*-tests were performed to compare the different groups. A *p*-value of less than 0.05 was considered statistically significant. The statistical analyses were performed using the SPSS software 17.0 (SPSS Inc., Chicago, IL, USA) for Windows. All of the linear regression analyses described in this paper were processed using version 6.0 of the Origin professional software. 

## 4. Conclusions and Perspective 

Myricetin-3-*O*-galactoside (M3OGa) and myricetin aglycone may trap free radicals via a chain reaction comprising of a propagation step and a termination step. The 3-*O*-galactosylation process can limit the redox-dependent antioxidant pathways at the propagation step; this makes myricetin-3-*O*-galactoside an inferior antioxidant compared to its aglycone. Nevertheless, both can afford similar RAF products at the termination step. This effect of 3-*O*-galactosylation confirmed that the antioxidative activities of phytophenols are differentiated at the propagation step rather than the termination step.

## Figures and Tables

**Figure 1 molecules-24-02769-f001:**
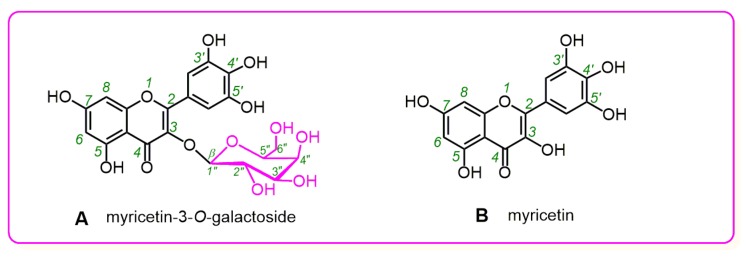
Structures of myricetin-3-*O*-galactoside (M3OGa) (**A**) and myricetin (**B**).

**Figure 2 molecules-24-02769-f002:**
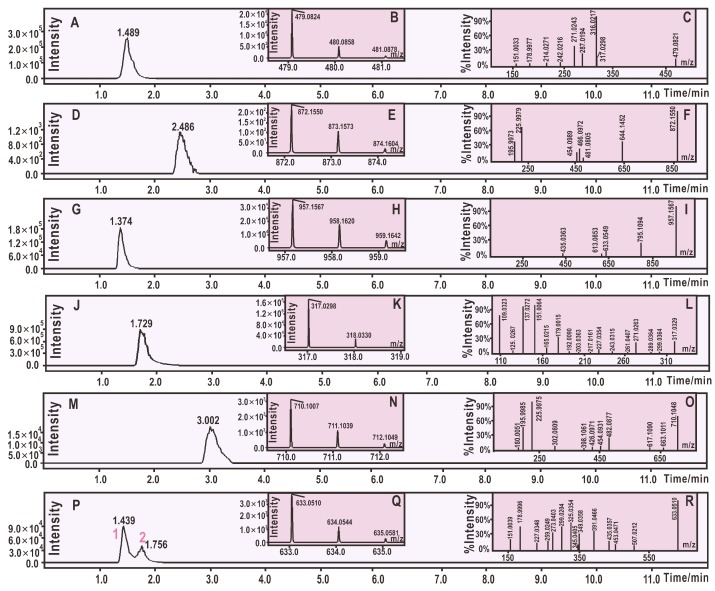
Typical results of the UPLC-ESI-Q-TOF-MS analysis: (**A**) Chromatogram of myricetin-3-*O*-galactoside (M3OGa) when the formula, [C_21_H_20_O_13_-H]^−^, was extracted; (**B**) primary MS spectra of M3OGa; (**C**) secondary MS spectra of M3OGa; (**D**) chromatogram of the radical adduct formation (RAF) product of M3OGa–DPPH when the formula, [C_39_H_31_N_5_O_19_-H]^−^, was extracted; (**E**) primary MS spectra of the RAF product of M3OGa–DPPH; (**F**) secondary MS spectra of the RAF product of M3OGa–DPPH; (**G**) chromatogram of possible dimeric products of M3OGa when the formula, [C_42_H_38_O_26_-H]^−^, was extracted; (**H**) primary MS spectra of possible dimeric products of M3OGa; (**I**) secondary MS spectra of the RAF product of the dimeric products of M3OGa; (**J**) chromatogram of myricetin when the formula, [C_15_H_10_O_8_-H]^−^, was extracted; (**K**) primary MS spectra of myricetin; (**L**) secondary MS spectra of myricetin; (**M**) chromatogram of the RAF product of myricetin–DPPH when the formula, [C_33_H_21_N_5_O_14_-H]^−^, was extracted; (**N**) primary MS spectra of the RAF product of myricetin–DPPH; (**O**) secondary MS spectra of the RAF product of myricetin–DPPH; (**P**) chromatogram of possible dimeric products of M3OGa when the formula, [C_30_H_18_O_16_-H]^−^, was extracted; (**Q**) primary MS spectra of possible dimeric products of myricetin; (**R**) secondary MS spectra of the RAF product of the dimeric products of myricetin.

**Figure 3 molecules-24-02769-f003:**
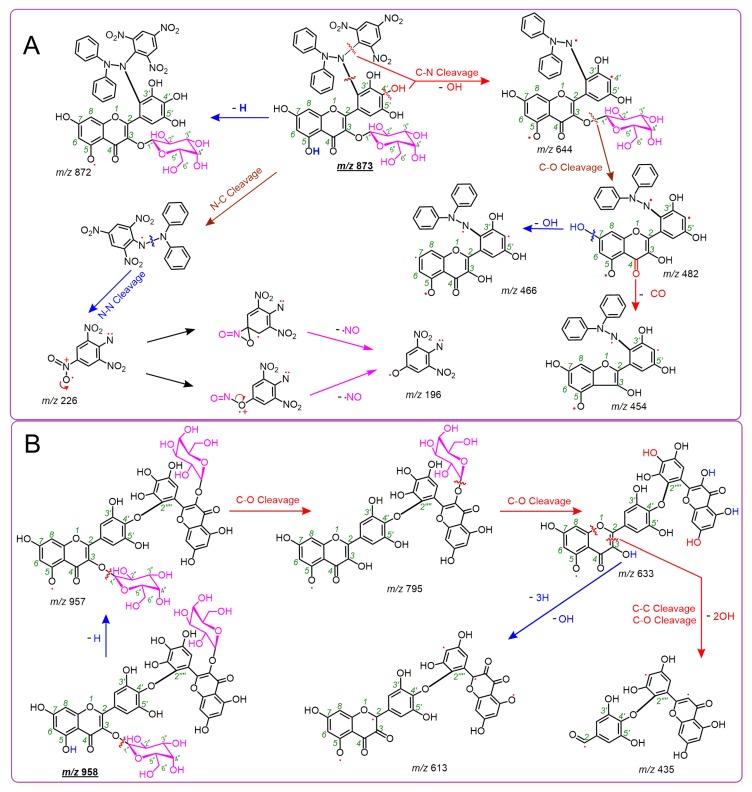
Proposed MS elucidations of the RAF reaction products between M3OGa and the DPPH^•^ radical. (**A**) M3OGa–DPPH adduct; (**B**) M3OGa–M3OGa dimer (the MS spectra were in the negative ion mode. The accurate *m*/*z* values are simply expressed as integers. Other linking sites between the M3OGa and DPPH moieties and other reasonable cleavages should not be excluded in the MS elucidation).

**Figure 4 molecules-24-02769-f004:**
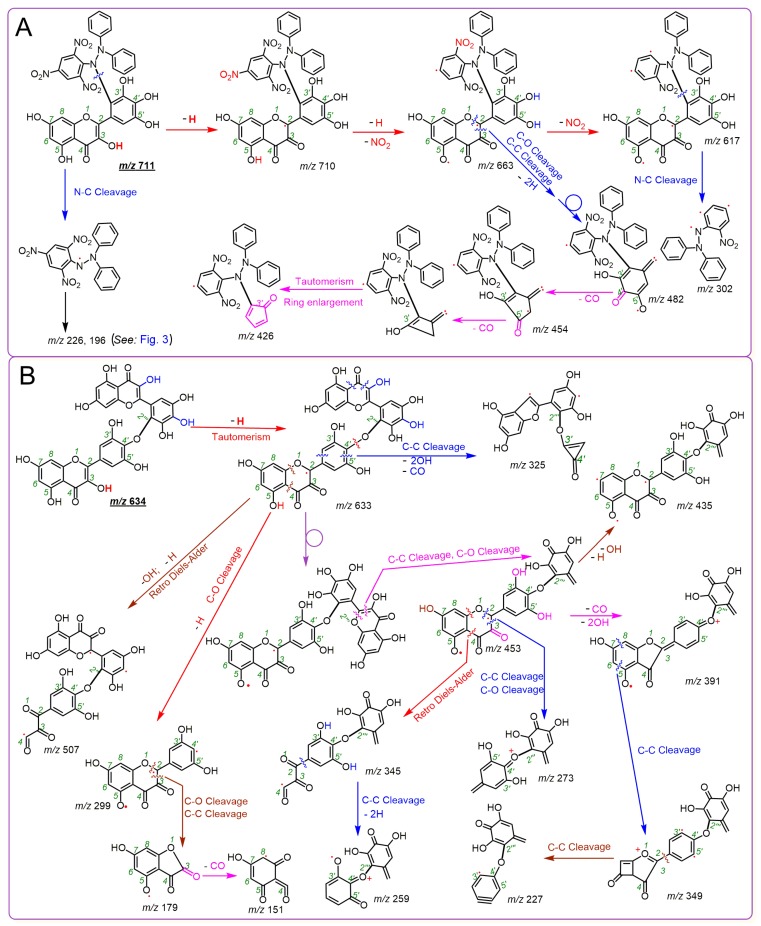
Proposed MS elucidations of the RAF reaction products between myricetin and the DPPH^•^ radical. (**A**) Myricetin–DPPH adduct; (**B**) myricetin–myricetin dimer (the MS spectra were in the negative ion mode. The circle indicates σ bond rotation. The accurate *m*/*z* values are simply expressed as integers. Other linking sites between the myricetin and DPPH moieties and other reasonable cleavages should not be excluded in the MS elucidation).

**Figure 5 molecules-24-02769-f005:**
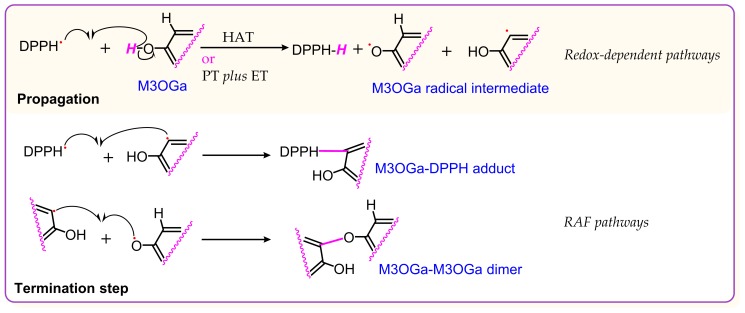
Proposed chain reaction of M3OGa trapping 2 mol DPPH^•^.

**Figure 6 molecules-24-02769-f006:**
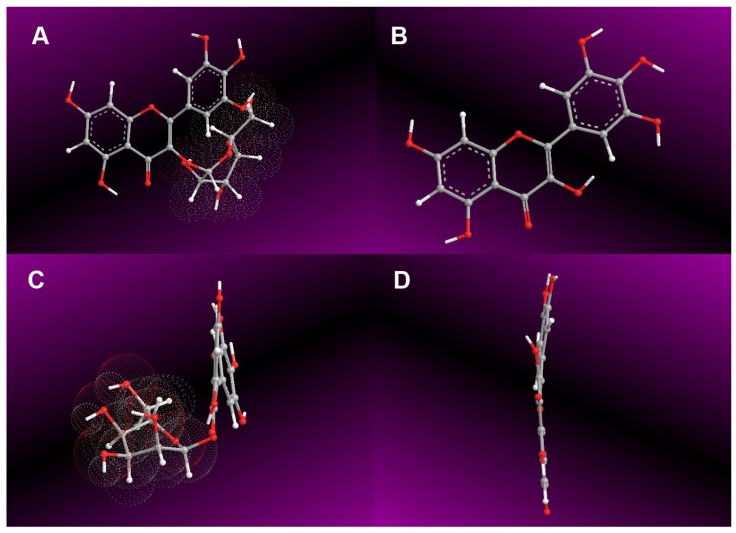
Preferential conformations of myricetin-3-*O*-galactoside (M3OGa) and myricetin. (**A**) Front view of M3OGa; (**B**) front view of myricetin; (**C**) right side view of M3OGa; (**D**) right side view of myricetin. The preferential conformation was analyzed using the Chem3D Pro14.0 program (PerkinElmer, Waltham, MA, USA).

**Table 1 molecules-24-02769-t001:** IC_50_ values (μM) of M3OGa and myricetin in the antioxidant spectrophotometric analyses.

Antioxidant Analyses	M3OGa	Myricetin	Trolox
DPPH^•^-trapping	12.9 ± 0.3 ^b^	10.7 ± 0.3 ^a^	26.4 ± 2.5
PTIO^•^-trapping (pH 4.5)	263.7 ± 3.5 ^b^	132.9 ± 5.1 ^a^	220.1 ± 4.6
PTIO^•^-trapping (pH 7.4)	131.2 ± 5.1 ^b^	81.5 ± 2.4 ^a^	142.9 ± 5.0
^•^O_2_^−^-trapping	88.9 ± 7.2 ^b^	73.3 ± 1.9 ^a^	2777.5 ± 35.3

The IC_50_ value (in μM) was defined as the final concentration of 50% radical inhibition or relative reducing power, determined by linear regression analysis and expressed as the mean ± SD (*n* = 3). The linear regression was analyzed using version 6.0 of the Origin professional software. The IC_50_ values with different superscripts (a or b), between M3OGa and myricetin, are significantly different (*p* < 0. 05). Trolox is the positive control. The dose response curves are listed in Appendix A.

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
