# Peer review of "Comparative Analysis of Radical Adduct Formation (RAF) Products and Antioxidant Pathways between Myricetin-3-O-Galactoside and Myricetin Aglycone"

_molecules, 2019, doi:10.3390/molecules24152769_

Round 1
Reviewer 1 Report
The manuscript described the analysis of radical adduct formation (RAF) and the antioxidant pathway between myricetin-3-O-galactoside and the respective aglycone.
The method of analysis was based on UPLC-ESI-Q-TOF-MS
The work can be improved by separating the sections Results and Discussion: Table 1 and figure 2 are the results and the other figures showed the proposed mechanism based on m/z.
Reviewer 2 Report
In this work, in a comparative study, the final products of the interaction of M3OGa and myricetin aglycone with a commercial radical compound were analyzed using ultra-performance liquid chromatography coupled with mass spectrometry to test the possibility of radical adduct formation.
The authors worked methodically and the manuscript is well structured.
The literature is sufficiently updated.
The Results and Discussion section is correctly written. Thanks to the methodical presentation of the results obtained from the work, correct conclusions were drawn. The results provided interesting information on the antioxidant chemistry of all types of phytophenols. I have no comments on this part of the work.
Reviewer 3 Report
The work is interesting and sound. It describes in details the potential pathways of interaction of polyphenolic antioxidants with free radical exemplified as DPPH radical.
I have only a few suggestions:
lines 87-90: this sentence is not clear and lacks verb
lines 115-120: this information is quite basic, no need to show in the manuscript
Figure 3A: the fragmentation in negative mode should follow molecular ion of [M-H], the molecule 644 miss radical dot at C4', please correct the fragmentation including initail loss of [H]
lines 131-134: this information is quite basic, no need to show in the manuscript
Figure 4B: molecular ion in negative mode is always missing H, hence it should give fragments only without H
Suppl data 3: Trolox activity curves should also be shown on a graphs
